# Damselfish face climate change: Impact of temperature and habitat structure on agonistic behavior

**Thalles da Silva-Pinto**[1], **Mayara Moura Silveira**[1], **Jéssica Ferreira de Souza**[1], **Ana Luisa Pires Moreira**[1], **Edson Aparecido Vieira**[2], **Guilherme Ortigara Longo**[2], **Ana Carolina Luchiari**[1]*

1 Laboratório de Peixes, Departamento de Fisiologia e Comportamento, Centro de Biociências, Universidade Federal do Rio Grande do Norte, Natal, Brazil, 2 Laboratório de Ecologia Marinha, Departamento de Oceanografia e Limnologia, Centro de Biociências, Universidade Federal do Rio Grande do Norte, Natal, Brazil

* analuchiari@yahoo.com.br

## Abstract

Oceans absorb a huge part of the atmospheric heat, leading to the rise in water temperature. Reefs are among the most affected ecosystems, where the complex behavioral repertoire of fishes is usually an indicator of environmental impacts. Here, we examined whether temperature (28 and 34˚C) and habitat complexity (high and low) interact to affect the agonistic behavior (mirror test) of the dusky damselfish (*Stegastes fuscus*), a key species in Brazilian reefs because of its gardening capacity and territorial behavior. Higher temperatures altered basal behavior in both high and low-complexity conditions. Fish kept at 28˚C under the high-complexity condition were more aggressive than those at a higher temperature (34˚C) and in a low-complexity condition, which also exhibited lower dispersion. Our data show that changes in behavior of coral reef fish is associated to fluctuations in environmental conditions. Thus, it is important to implement management or conservation strategies that could mitigate global change effects.

## 1. Introduction

Global warming has been singled out as one of the most devastating effects of human activities [1]., especially for oceans, which absorb around 90% of the atmospheric heat. For water-breathing ectothermic, physiological functioning depends on the thermal condition [2,3]. The rise in water temperature directly affects fishes' metabolism, increasing respiration rate, nutritional requirements, and other physiological and behavioral responses such as reproduction and immunological defense [4–8]. Moreover, for the reef fishes the impact is exacerbated because algae and coral that make up reef's structural foundation are even more sensitive to temperature rise. The three-dimensional structure and food source offered by the coral communities favor the establishment of hundreds of species, and the immediate impact of warming will be loss of diversity and changes in fish community composition [9,10].

**Data Availability Statement:** All relevant data are within the manuscript and its Supporting Information files.

**Funding:** MMS, JFdS, ALPM and EAV were financed in part by the Coordination for the Improvement of Higher Education Personnel (CAPES). The funders had no role in study design, data collection and analysis, decision to publish, or preparation of the manuscript.

**Competing interests:** The authors have declared that no competing interests exist.

Reef fishes exhibit a complex behavioral repertoire, and subtle changes in behavior are usually associated to fluctuations in environmental conditions [7]. Thus, reef fishes are important organisms in determining how environmental changes can modulate behavior and help us understand their potential influence on reef ecosystems. Environmental changes were shown to affect fish cognition [11], personality [12], reproduction patterns [13], social interactions, migration and even species diversity of reef dwellers [14].

Some reef-fish populations are sensitive to slight increases in water temperature [15], while others tolerate higher temperature variations [16,17]. The dusky damselfish (*Stegastes fuscus*) occurs along almost the entire Brazilian coast (from 5˚S to 27˚S), featuring a wide range of thermal gradients. Studies on a related species from the Pacific (*Chromis atripectoralis*) indicated that the thermal optimum is around 1˚C above its regional maximum summer temperature (30˚C; [17]). Damselfishes play an important ecological role by affecting the structure of benthic and coralline communities [18–20] and controlling algal diversity [21]. These species are largely territorial, exhibit aggressive behavior toward other herbivorous species and contribute to energy and nutrient transfer in reef environments, as a result of their gardening ability [22].

If global warming projections for the end of this century materialize, ocean water temperatures may increase 2–4˚C on average, and important conditions for preserving marine life will be affected [23,24], including the structural complexity of reefs [25]. In this study we evaluated whether water temperature and structural complexity of the habitat affect mobility patterns, tank occupation, and behavioral profile of the damselfish *Stegastes fuscus*. For this, we subjected the animals to classic mirror test and observed if animals kept at high temperature and barren conditions (mimicking the worst forecast scenario for the future) present significant changes in behavior when compared to fish kept at natural temperature and enriched or barren habitat. As increase in temperature raises the metabolic rate of fish and promotes direct influences in behavior, we suggest that natural aggressive behavior of *S. fuscus* would be affected.

## 2. Materials and methods

### 2.1 Animal sampling and holding conditions

Animals were collected from Pirambúzios beach (6˚03'25"S and 35˚05'53"W), Nísia Floresta, Rio Grande do Norte state, Northeastern Brazil, as authorized by the Brazilian Institute of Environment and Natural Resources (IBAMA License Number 62318-1/2018). The tide pools that form at this beach serve as a refuge for various marine communities [26]. The average maximum coastal water summer temperature is 30˚C [27], but reaches 36˚C in the tide pools (critical temperature occurring only at low tide and for short periods of time), salinity remains between 36 and 40ppt, and pH is around 8.0 [28].

Dusky damselfish (*S. fuscus*) were collected from the tide pools in two sampling moments (average size and weight of 8.99 ± 1.04 cm, 18.83 ± 4.29 g and 8.27 ± 0.80 cm, 13.41 ± 2.99 g at the first and second capture, respectively) using a cast net (3m diameter, 10mm mesh size). Fish were immediately stored in 30-L containers with seawater and air stones to maintain oxygen level. Next, they were taken to the laboratory and placed in glass tanks (33 x 30 x 30 cm; 25L) at the Fish Vivarium, Department of Physiology and Behavior, Federal University of Rio Grande do Norte. Saltwater was previously prepared (Red Sea Salt, Red Sea, Houston, USA) and the tanks filled. Salinity was maintained at 36ppt, and a 12:12H light:dark cycle was established.

Every 12 tanks formed a closed recirculating system, in which water was kept aerated and filtered (mechanical, chemical and biological filters) and maintained at a controlled

temperature by a thermostat. Fish were individually held in isolated tanks to avoid physical confrontation and damages. One system (12 tanks) was kept at 28°C, the average water temperature on the Brazilian coastal reefs where animals were sampled [27], and the other (12 tanks) at 34°C, the expected warming of tropical oceans projected to occur by the end of this century [24]. Since 34°C is considered the long-term thermal limit for several species of reef fish [17], it represents the worst global warming scenario. To reach 34°C, tanks at 28°C were subjected to a 0.5°C temperature increase every 2 hours for 24h. Tanks were also enriched or kept clean to provide a complex or barren habitat, respectively. The high-complexity habitat consisted of covering the walls and bottom of the tank with wallpaper simulating marine gravel substrate, and including a shelter (6 x 6 x 15 cm hollowed brick) and plastic plants in the tanks. The barren (low-complexity) habitat contained none of the aforementioned items and the tank was kept completely clean. Thus, four groups were formed: "complex habitat at 28°C" (28C group–n = 12); "complex habitat at 34°C" (34C group–n = 06); "barren habitat at 28°C" (28B group–n = 9); and "barren habitat at 34°C" (34B –n = 9). Fish were kept in these conditions for 1 month before the behavioral tests. They were fed twice a day *ad libitum* with frozen *Artemia salina*, shrimp paste and dried food pellets (algae-based tetra marine salt granules). When any type of disease/injury was observed during the 30- day period or when fish stopped feeding for more than 5 days, fish were excluded from the behavioral test, resulting in groups with different sample sizes. A total of twelve fish were excluded from the tests. Following the research data collection, all animals used were euthanized using clove oil anesthetic. All animal procedures were authorized by the Animal Ethics Committee of the Federal University of Rio Grande do Norte (CEUA 100.12/2018).

## 2.2 Behavioral tests

The experimental tanks (40 x 20 x 25cm, 15L) were filled with water under the same conditions and temperature as the stocking systems where each group was maintained, and an air stone provided constant aeration. The tanks were covered with white paper to prevent the fish from having any contact with the outside environment, but the right and the front walls remained uncovered for experimental purposes. A white partition was placed in front of the right wall, but could be removed when needed, allowing the fish to see a mirror positioned at 45° [28–31]. In this position, one corner of the tank (Q1) was closer to the mirror than the other (Q2). On the opposite sides of the mirror (Q3 and Q4) were areas of less interactivity with the mirror image, where less responsive animals were expected to remain longer (Fig 1). The uncovered front wall allowed a camera (SONY® DCR-SX45) to record fish behavior (camera positioned 50 cm from the tank), while another camera (SONY® DCR-SX45) was placed 1m above the tank in order to record fish movements. Fish were not fed during behavioral tests.

Fish from the 4 stocking conditions (28C, 34C, 28B, 34B) were individually placed in the center of experimental tank, and after a 2- min acclimation period, behavior was recorded from the overhead and frontal cameras. During the first 5 min of recording, the white partition prevented the fish from seeing the mirror. The partition was then removed and fish behavior was recorded for another 5 min. The animals were then returned to the stock tanks.

Behavior recorded by the frontal camera was visually tracked to identify aggressive displays directed toward the mirror. The number and type of displays were quantified. *Attacks* where considered when the fish approached the mirror from the front or the side quickly and/or successively, opening its mouth and trying to bite the side of the tank in contact with the mirror, and *Threats* when it erected its dorsal, pelvic and anal fins close to the mirror. *Immobility* was considered when the fish remained still for 2s or more, *Vigilance* when it moved through the tank fins down, and *Substrate nibbling* when it bit any possible particle at the bottom of the

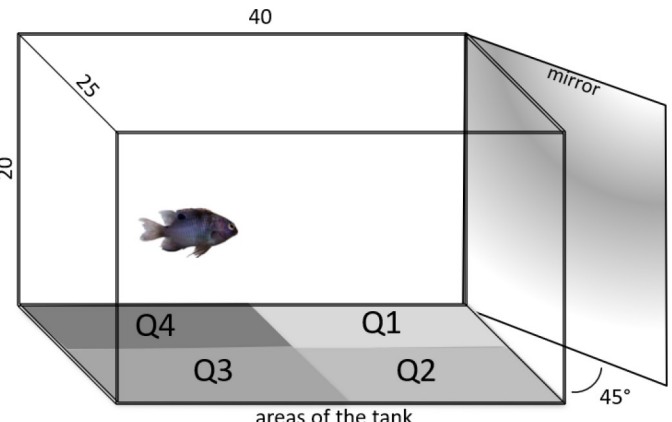

**Fig 1. Schematic overview of the tank (15L, 40x20x25 cm) used for the mirror test.** Walls were covered with opaque white film, except the front and mirror walls. The wall that allowed mirror view was covered for the first 5 min of behavioral recording (basal behavior). The cover was then removed and the fish had access to its own image for 5 min. Behavior was recorded from the front to analyze aggression, and from above to assess fish distribution in the tank. Remaining longer in quadrant 1 (Q1) indicates close contact with the mirror image, followed by Q2, while lingering in Q3 and Q4 suggests greater distance from the mirror image.

tank (no food was offered, thus it was recorded as a typical foraging behavior observed in nature). Locomotion was recorded from the overhead camera and analyzed using tracking software developed in MatLab [32]. The overhead view made it possible to divide the bottom of the tank into 4 quadrants: Q1 was the closest to the mirror, Q2 the side area where the mirror offered a more distant view of the image, and Q3 and Q4 were the back areas (Fig 1; see video at https://www.youtube.com/watch?v=wz_aOkiunOA&feature=youtu.be). The time spent in each quadrant and average swimming speed were measured. For speed, calculations are performed on a series of frames to produce quantified measurements of the animal behavior. It is known the position of the animals for each frame of the video, and the number of frames per second. Thus, the series of frames is analyzed, and the number of frames changed in a certain time is used to estimate the animal's movement.

## 2.3 Statistical analysis

All comparisons were performed through independent analyses before and after mirror exposure. We decided not to use 'time' as a factor due to the dependence between two periods and because analysis of time spent in each quadrant (see details below) would result in two dependent factors in the same analysis ('time' and 'quadrant').

The effects on mobility were evaluated using swimming velocity and immobility time. Each variable was compared separately between 'temperature regime' (fixed, '28˚C' and '34˚C') and 'habitat complexity condition' (fixed, 'complex' and 'barren'), and the respective interaction, applying two-way ANOVA. Data were checked for normality (Komogorov-Smirnov test) and homoscedasticity (Levene test), and when a divergence was observed (swimming velocity and immobility time after mirror exposure) square-root transformation was applied. For significant sources of variation, *posthoc* pairwise comparisons were evaluated using the Student-Newman-Keuls test.

The effects on aggressive behavior were assessed considering the time spent in each quadrant, related to mirror position, as a proxy of aggressiveness level (see methods). Since the time spent in one quadrant is dependent on the others, repeated measures ANOVA was performed for each of them, with 'temperature regime' ('28˚C' and '34˚C') and 'habitat complexity'

('complex' and 'barren') as fixed factors, and 'quadrant' (Q1, Q2, Q3 and Q4) as the repeated measure. Data were square-root transformed and Greenhouse-Geisser correction was applied when epsilon was lower than 0.75 [33]. For significant sources of variation, the differences were highlighted by applying descriptive analysis [34].

Behavioral effects were evaluated using a multivariate approach. After the data on percentage of behaviors were square-root transformed, they were used to build a resemblance matrix with Bray-Curtis distance. To test homogeneous dispersion, data were assessed using the PERMDISP procedure (permuted dispersion, which tests for homogeneity of dispersions). Following these procedures, PERMANOVA with 999 permutations [35] was carried out using the same model applied in univariate comparisons for swimming velocity and immobility time. For significant sources of variations, pairwise comparisons were performed and SIMPER analyses conducted to highlight the behaviors that most contributed to these differences.

The univariate analyses (two-way and repeated measures ANOVA tests) were performed in the software Systat 12 and the multivariate procedures (PERMANOVA and SIMPER tests) in the software Primer 6 with PERMANOVA add-on.

## 3. Results

Swimming velocity varied between habitat complexity depending on the temperature regime, both before and after mirror exposure (Table 1 and S1 Data). The differences between habitat complexity occurred only at 34 degrees. Fish from the complex habitat decreased velocity before mirror exposure and those from the barren habitat increased it after the same exposure (Table 1 and Fig 2). For immobility time, we observed an effect only after mirror exposure, where fish from barren tanks showed a decrease at 34°C (Table 1 and Fig 2).

The time spent in different quadrants under the two temperature regimes depended on habitat complexity, both before and after mirror exposure (Table 2). Before exposure at 28°C, the fishes spent more time in quadrant 4 (lower aggressiveness), with a greater difference observed when the habitat was barren (28C versus 28B). At 34°C, although the fishes also spent more time in quadrant 4, the opposite was observed, with a larger difference when the

**Table 1. Two-way ANOVA to compare 'Swimming velocity' and 'Immobility time' between temperature (28 and 34°C) and habitat structure (complex and barren) in the dusky damselfish before and after mirror exposure.**

| | | Swimming velocity | | | | | |
|---|---|---|---|---|---|---|---|
| | | **Before mirror** *(KS–p = 0.128; L–p = 0.718)* | | | **After mirror** *(KS–p = 0.061; L–p = 0.493)* | | |
| Source | DF | MS | F | P | MS | F | P |
| Temp. | 1 | 0.55 | 0.14 | 0.714 | 1.14 | 6.32 | **0.017** |
| Compl. | 1 | 2.14 | 0.53 | 0.471 | 2.39 | 13.20 | **0.001** |
| T x C | 1 | 33.60 | 8.34 | **0.007** | 2.81 | 15.55 | **< 0.001** |
| Error | 32 | 4.03 | | | 0.18 | | |
| | | **Before mirror** *(KS–p = 0.457; L–p = 0.141)* | | | **After mirror** *(KS–p = 0.125; L–p = 0.373)* | | |
| Source | DF | MS | F | P | MS | F | P |
| Temp. | 1 | 18.67 | 0.01 | 0.941 | 19.73 | 4.26 | **0.047** |
| Compl. | 1 | 4.08 | 0.00 | 0.973 | 24.43 | 5.27 | **0.028** |
| T x C | 1 | 12892.76 | 3.80 | 0.060 | 33.49 | 7.23 | **0.011** |
| Error | 32 | 108674.99 | | | 4.63 | | |

Square root-transformed data were used for swimming velocity after mirror exposure in order to achieve ANOVA assumptions. Bold p-values correspond to significant effects. KS–Kolmogorov-Smirnov test for normality; L–Levene test for homoscedasticity; Temp./T–temperature; Compl./C–complexity of the habitat. DF = degrees of freedom, MS = mean squared

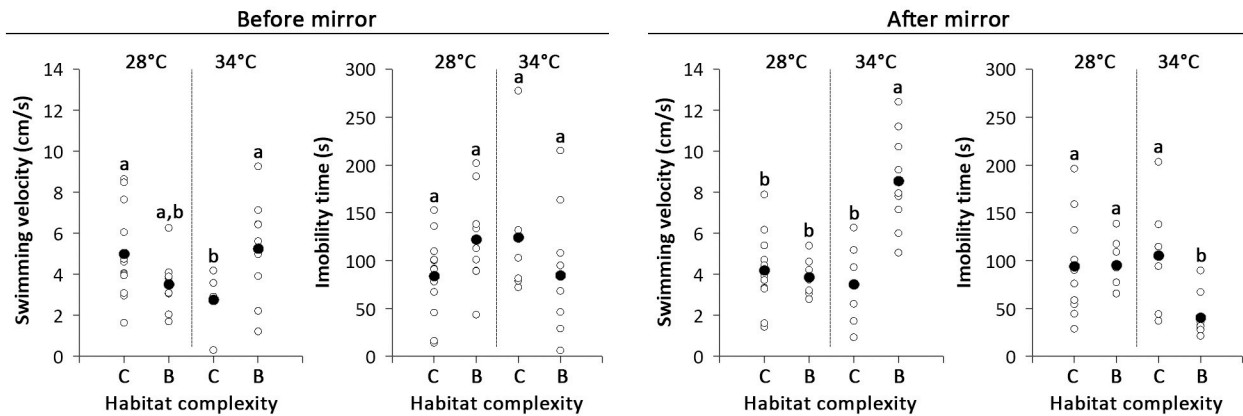

**Fig 2. Effects of temperature (28 and 34˚C) and habitat structure (complex and barren) on dusky damselfish mobility (swimming velocity and immobility time) before and after mirror exposure.** Combinations of temperature and habitat structure with the same letter above the bars are not statistically significant (p>0.05 in Two-Way ANOVA tests) (For details, see Table 1). Data correspond to average values ±SE.

habitat was structurally more complex (Fig 3). After mirror exposure, fishes remained longer in quadrant 1 (*i.e.* more aggressive) under both temperature regimes. However, fish at 28˚C showed a greater behavioral change (from quadrant 4 to quadrant 1), while fish at 34˚C occupied the other quadrants (less aggressive when compared to quadrant 1). Also, fish from complex habitat remained longer in quadrant 1 than fish from barren habitat (Fig 3).

Aggressive and locomotor behavior differed due to the temperature before mirror exposure and the interactive effect of temperature and habitat complexity after mirror exposure (Two-Way PERMANOVA, Table 3). Before mirror exposure, more threat behavior occurred at 34˚C and more vigilance, immobility and feeding at 28˚C (Fig 4 and Table 4). However, there was a trend to a habitat complexity effect at 34˚C, resulting in more threat behavior in the barren condition and more vigilance in its complex counterpart (Fig 4 and Table 4). After mirror exposure, attack behavior became more frequent (almost absent before mirror exposure) and an integrative effect between temperature regime and habitat complexity was observed (Fig 4

**Table 2. Repeated Measures ANOVA for comparisons of time spent in each quadrant considering the different temperatures (28 and 34˚C) and environments (complex and barren) in the dusky damselfish before and after mirror exposure.**

| | | | Before mirror (ε – 0.716) | | | After mirror (ε – 0.689) | | |
|---|---|---|---|---|---|---|---|---|
| Source | DF | MS | F | P | MS | F | P |
| *Between subjects* | | | | | | | | |
| Temp. | 1 | 1.17 | 0.32 | 0.575 | 1.86 | 0.76 | 0.390 |
| Compl. | 1 | 3.19 | 0.88 | 0.357 | 8.80 | 3.61 | 0.067 |
| T x C | 1 | 10.50 | 2.88 | 0.100 | 0.60 | 0.25 | 0.624 |
| Error | 32 | 3.65 | | | | | |
| *Within subjects* | | | | | | | | |
| Quadr. | 3 | 181.44 | 13.90 | < **0.001** | 166.46 | 16.27 | < **0.001** |
| Q x T | 3 | 0.92 | 0.07 | 0.942 | 27.26 | 2.67 | 0.075 |
| Q x C | 3 | 2.90 | 0.22 | 0.817 | 1.97 | 0.19 | 0.832 |
| Q x T X C | 3 | 47.50 | 3.64 | **0.029** | 33.06 | 3.23 | **0.044** |
| Error | 96 | 13.05 | | | 10.23 | | |

Data were square-root transformed and p-values were subjected to Greenhouse-Geisser correction. Bold p-values correspond to significant effects. Temp./T–temperature; Compl./C–complexity of the habitat; Quadr/Q–quadrant. DF = degrees of freedom. MS = mean squared

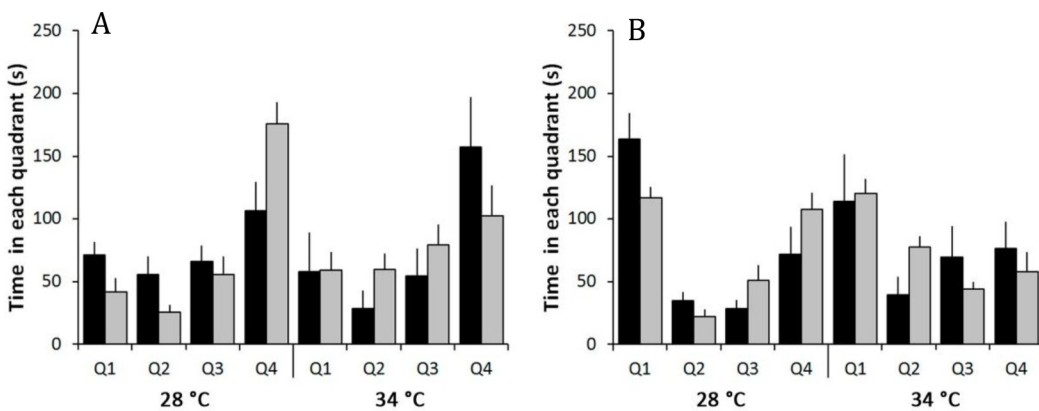

**Fig 3. Effects of temperature (28 and 34°C) and habitat structure (complex = black bars and barren = gray bars) on dusky damselfish dispersion in the tank (time spent in each quadrant) before (graph A) and after (graph B) mirror exposure (p>0.05 in Repeated Measures ANOVA tests) (For details, see Table 2).** Data correspond to average values ±SE; no pairwise comparisons were performed since this was analyzed using Repeated Measures ANOVA.

and Table 3). Overall, there were more attacks at 28°C when compared to 34°C, but habitat complexity played a larger role at 34°C, with more attacks and threats when the habitat was barren and more immobility and vigilance when it was complex (Fig 4 and Table 4).

## 4. Discussion

We observed that changes in temperature and complexity affect mobility and aggressiveness in the dusky damselfish (*S. fuscus*). An increase in water temperature from 28 to 34°C (mimicking global warming predictions) and a decrease in habitat complexity (mimicking the loss of complexity due to coral mortality predicted in global warming scenarios) increased motor activity and decreased the emission of typical agonistic displays. This result suggests that environmental temperature and structure affect the natural behavioral repertoire of *S. fuscus* and thus, the strength of interspecific competition may be disturbed and affect habitat use and the interactions in the reef community. Although the species shows high behavioral plasticity [36], it may require longer to properly cope with drastic environmental changes.

It is forecasted that the rate of warming will accelerate in the near future [37–39]. Thermal stress, one of the leading direct climate-related threats to reef ecosystems [40], may result in increased frequency and intensity of coral bleaching [41], algae population decline [42] and dispersion of reef species [43,44]. Warming may lead to increased metabolic activity in fishes, potentially causing cardiac and ventilatory overload [15] and consequent reduction in the

**Table 3. Two-way PERMANOVA comparing the set of behaviors under different temperatures (28 and 34°C) and habitat structures (complex and barren) in the dusky damselfish before and after the mirror exposure.**

| Source | DF | Before mirror (PERMDISP–p = 0.930) | | | After mirror (PERMDISP–p = 0.057) | | |
| | | MS | Pseudo-F | P-perm | MS | Pseudo-F | P-perm |
|---|---|---|---|---|---|---|---|
| Temp. | 1 | 2997.4 | 4.77 | **0.007** | 1166.8 | 2.91 | 0.060 |
| Compl. | 1 | 2065.3 | 3.29 | 0.051 | 636.8 | 1.59 | 0.205 |
| T x C | 1 | 1884.9 | 3.00 | 0.069 | 2258.5 | 5.64 | **0.009** |
| Error | 32 | 628.42 | | | 400.5 | | |

Data were square-root transformed in order to better achieve homogeneous dispersion. Bold p-values correspond to significant effects. Temp./T–temperature; Compl./ C–complexity of the habitat. DF = degrees of freedom, MS = mean squared, PERMDISP = permuted dispersion.

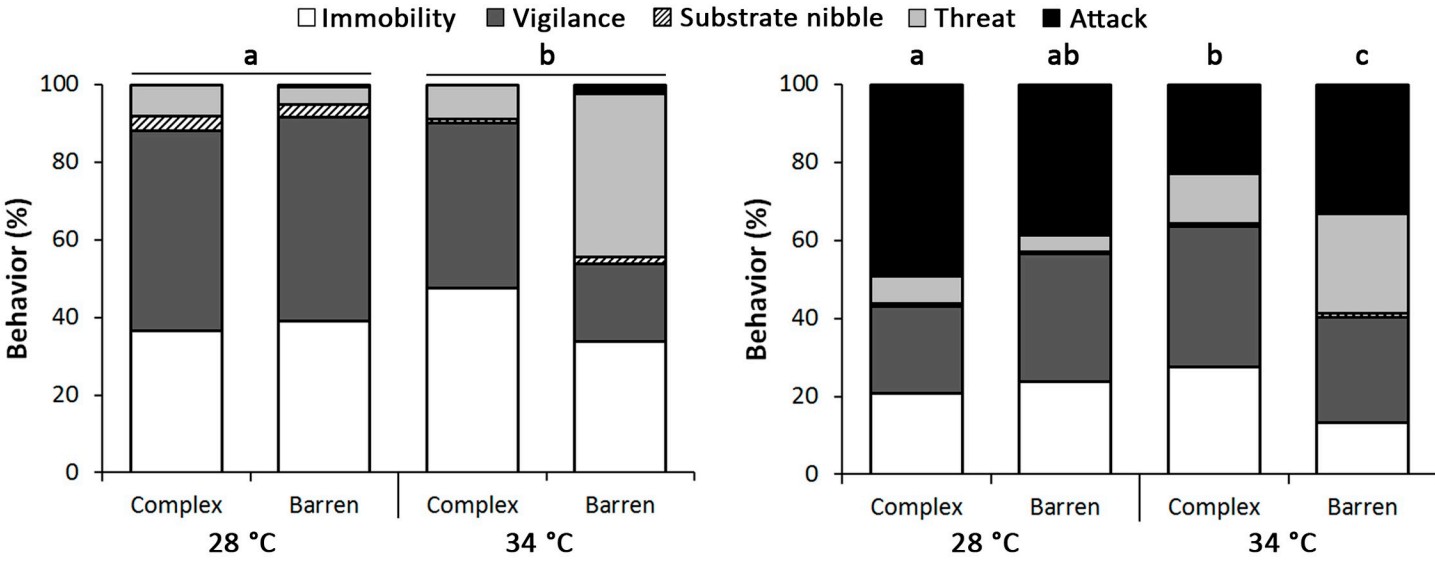

**Fig 4. Effects of temperature (28 and 34°C) and habitat structure (complex and barren) on dusky damselfish behavior before and after mirror exposure.** Combinations of temperature and complexity with the same letter above the bars are not statistically significant after pairwise comparisons (p>0.05 in pairwise comparisons using permutation tests) (For details, see Tables 3 and 4).

animal's aerobic scope [45]. Thermal stress was already shown to cause behavioral changes that are usually followed by reduced ability to exploit resources [46], decreased immunity response [47,48], reproductive losses [6,49], and failure to recognize environmental cues [50,51]. A recent study on butterflyfishes showed reduced aggression due to reef's complexity loss [4].

The dusky damselfish appears to tolerate some temperature variation: it can withstand acute temperature increase caused by seawater entrapped in small tide pools that can reach 36°C (personal observation). However, their tolerance time-range seems to be narrow (i.e., only 4–6 hours of the low tide). In this scenario, fish seem to reduce locomotion and the percentage of time invested in agonistic behaviors until the flow returns to higher levels, and the animals have access to the open sea. In contrast, the present study imposed a more extended period of high temperature (30 days), and *S. fuscus* tolerance zone may have been exceeded to enter the resistance zone. Warming caused a decrease in costly displays (threats, attacks) that were replaced by more economical ones, such as vigilance. Although fish spent more time close to the mirror (after mirror exposure), showing that warming did not affect the ability to

**Table 4. SIMPER results for the behaviors that most contributed to the differences observed after pairwise comparison for significant effects obtained in PERMANOVA.**

| Behavior | Before mirror | After mirror | | | |
|---|---|---|---|---|---|
| | 28 *vs.* 34 | 28C *vs.* 34C | 28C *vs.* 34B | 28B *vs.* 34B | 34C *vs.* 34B |
| Immobility | 3rd (28) | 3rd (34C) | 4th (28C) | 2nd (28B) | 4th (34C) |
| Vigilance | 2nd (28) | 4th (34C) | 2nd (28C) | 3rd (28B) | 3rd (34C) |
| Feeding | 4th (28) | | | | |
| Threat | 1st (34) | 2nd (28C) | 1st (34B) | 1st (34B) | 1st (34B) |
| Attack | | 1st (28C) | 3rd (28C) | 4th (28B) | 2nd (34B) |

For each comparison between two conditions, behaviors were ranked in terms of contribution. The code in parentheses indicates the condition in which the behavior was more frequent: 28°C, 34°C, Complex–C or Barren—B.

notice a conspecific invader, it changed the way fish interact with the intruder. Thus, increased water temperatures seem to affect the behavioral trade-off between signaling and fighting against the intruder, a result that is consistent with our prediction.

The locomotor patterns observed in *S. fuscus* did not indicate loss of swimming velocity (Fig 2), as found in other species of damselfish [52–55]. The decreased aggression observed in increased water temperature may be related to nutritional deficit following temperature rise and complexity loss, as an economic strategy to cope with increased metabolism [4, 56,57]. Therefore, the ecological relations of the damselfish with the coral reef community would be compromised.

We observed that fish kept in the barren environment exhibited more aggressive displays than those in a complex environment, similar to what occurs with salmonids [58]. Overall levels of aggression tend to be higher in less complex environments [12], given that more aggressive animals have preferential access to resources [59] or more mating opportunities [60]. Thus, it seems that barren environments, where there are fewer habitat sharing options, lead to a single solution: increased aggressiveness to guarantee territory, even if it involves more stress and higher energy costs.

Although the combination of high temperature and barren condition was the worst for *S. fuscus*, the enrichment of the 34°C ambient seemed to compensate thermic stress, as was suggested by Goldenberg et al. [61]. Other authors have shown that enrichment increases neurogenesis and decreases anxiety-like behavior in fish [62,63]. Several other studies have discussed the importance of environmental complexity, which seems to favor behavioral plasticity [64,65] and decline stress levels [66,67], in addition to affecting reef predation and competition rates [68,69]. The structural complexity of the habitat is related to ecologically diverse environments with high fish abundance [70], because a more complex environment provides elevated niche and resource variability [71], producing areas that can harbor more biological diversity and fish biomass when compared to less structurally complex environments [72,73]. Thus, habitat structure may ultimately change the behavioral pattern of species [74]. As such, the reef ecosystem complexity, which is considered of utmost importance for species diversity and richness [25,75–78], is being threatened by the global warming, and many species that occur exclusively on reefs are endangered.

The predicted global warming will affect species distribution and lead to profound changes in the three-dimensional structure of reefs [79]. In our study, we heated the tank water to the temperature predicted for the end of this century (34°C) and maintained it for one month, observing several behavioral effects on the dusky damselfish (*S. fuscus*) that may have cascading effects on reef community. Rather than overall effects on the reef, our results suggest ocean warming reduces the damselfish ability to maintain their territories and consequently control other species' growth by their gardening role. Losing territory size due to decreased aggressiveness is likely to scale up and affect interaction networks in the reefs (e.g., gardening, competition, predation, foraging, reproduction, navigability, and niche partitioning). Changes in behavior are only the tip of the iceberg of many and much more harmful changes, and strategies to mitigate global warming effects are urgently needed to prevent such a future adverse scenario.

## Supporting information

**S1 Data.**
(XLSX)

## Author Contributions

**Conceptualization:** Guilherme Ortigara Longo, Ana Carolina Luchiari.

**Formal analysis:** Thalles da Silva-Pinto, Edson Aparecido Vieira.

**Methodology:** Thalles da Silva-Pinto, Mayara Moura Silveira, Jéssica Ferreira de Souza, Ana Luisa Pires Moreira.

**Resources:** Ana Carolina Luchiari.

**Supervision:** Ana Carolina Luchiari.

**Writing – original draft:** Ana Carolina Luchiari.

**Writing – review & editing:** Thalles da Silva-Pinto, Mayara Moura Silveira, Jéssica Ferreira de Souza, Ana Luisa Pires Moreira, Guilherme Ortigara Longo, Ana Carolina Luchiari.

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
