## [Decision Letter · Decision Letter 0]

4 Nov 2019

PONE-D-19-24418

Damselfish face climate change: impact of temperature and habitat structure on agonistic behavior

PLOS ONE

Dear Dr. Luchiari,

Thank you for submitting your manuscript to PLOS ONE. After careful consideration, we feel that it has merit but does not fully meet PLOS ONE’s publication criteria as it currently stands. Therefore, we invite you to submit a revised version of the manuscript that addresses the points raised during the review process.

We would appreciate receiving your revised manuscript by January 2, 2020. To enhance the reproducibility of your results, we recommend that if applicable you deposit your laboratory protocols in protocols.io, where a protocol can be assigned its own identifier (DOI) such that it can be cited independently in the future. For instructions see: http://journals.plos.org/plosone/s/submission-guidelines#loc-laboratory-protocols

We look forward to receiving your revised manuscript.

Kind regards,

Hudson Tercio Pinheiro

Academic Editor

PLOS ONE

Journal Requirements:

2. In your Methods section, please include a comment about the state of the animals following this research. Were they released, euthanized or housed for use in further research? If any animals were sacrificed by the authors, please include the method of euthanasia and describe any efforts that were undertaken to reduce animal suffering.

Additional Editor Comments (if provided):

Dear Authors,

Based on the comments of the reviewers, the manuscript will require substantial revision and will need to be re-reviewed. I invite the authors to address each issue raised by the reviewers in a response letter, and submit a fully revised ms.

Kind regards,

Hudson T. Pinheiro

Reviewers' comments:

Reviewer's Responses to Questions

**Comments to the Author**

1. Is the manuscript technically sound, and do the data support the conclusions?

Reviewer #1: Partly

Reviewer #2: Yes

Reviewer #3: No

2. Has the statistical analysis been performed appropriately and rigorously? 

Reviewer #1: No

Reviewer #2: Yes

Reviewer #3: No

3. Have the authors made all data underlying the findings in their manuscript fully available?

Reviewer #1: Yes

Reviewer #2: No

Reviewer #3: Yes

4. Is the manuscript presented in an intelligible fashion and written in standard English?

Reviewer #1: Yes

Reviewer #2: No

Reviewer #3: No

5. Review Comments to the Author

Reviewer #1: General comments:

This study tries to address an important and timely topic regarding how warming might affect fish behaviour. However, the authors frame their work around only one behaviour (agonistic behaviour) for a single species, making the study not relevant to a broad public. Moreover, I have some concerns about a few passages of the text, which are hard to understand and flaws (manly on the figures) that could compromise the interpretation of the results. Additionally, the discussion and conclusion are based on motherhood statements that might not convince the readers. Given these problems, I wasn’t convinced that this manuscript presents compelling and generalisable messages for the readership of Plos One.

Specific comments:

Abstract:

Line 18: The author's statement “Due to the high anthropogenic impact of temperature, the pH, oxygen and structural complexity of many environments has changed.” sounds wrong. I would rephrase it. Warming is caused by the increase in atmospheric CO2 concentration, and the way that warming regulates pH could be somehow questionable, once that drops in pH (ocean acidification) is also directly associated with elevated atmospheric CO2 and primary production.

Line 24: This is unnecessary information “Fish behavior was recorded for 5 min before 25 and 5 min after mirror exposure.”

Line 28-30: Motherhood statement that does not move the field forward and also does not match with the study finds.

Introduction:

I found the introduction extremely hard to read, and sometimes sounds that the authors had copy paste statements in a shallow attempt to build a case. I would suggest to the authors to do a complete reconstruction of the introduction.

Line 35-58: The first two paragraphs could be easily mutated to a single paragraph.

Line 80-91: The hypothesis tested is based on assumptions not evaluated during the experiment which in my opinion is the biggest flaw in the study. Climate change is known for affecting species with different strength (some species may have suffered stronger physiological setbacks than others), such negative effects could be diminished or buffered by environmental complexity (see Goldenberg et al. 2018) and species interactions. Additionally, behaviour changes could be a simple and straightforward strategy to diminish physiological impairments or even take a chance to boost performance (Ferreira et al. 2018).

Materials and Methods:

Line 103: the sample size appears to be wrong by my calculations you collected 36 fish. Am I wrong?

Results:

First, due to low replication (sampling size vary from 6 to 12), I would appreciate seeing individual data points in the figures. Second, Figure 3, in my opinion, is not necessary. Figure 4, does not necessarily help since behaviour is not been affected by habitat complexity (left panel) and temperature (right panel) appears to have only a mild effect on fish behaviour, and as I said before, complexity buffer warming effects (Goldenberg et al. 2018).

I do have a question. During the behaviour test, the authors took feeding into account. However, is not clear in the experiment design how much food was available or how the authors made it available. Was it turf algae that grew in the experimental setup or it was food pallets? Was the food available in the same amount for all fish? Note that it could have serious implications on the results of the experiment. I would expect that fish in low complexity and higher temperatures would spend more time feeding (based on Metabolic Ecology Theory by Brown et al. 2004), or gardening (Ferreira et al. 2018, to avoid the overgrown of high nutritious algae by weedy ones in damselfish territories). Yet, the authors do not give any indication of such responses.

Discussion:

Here, I have the same problem as in the introduction. The majority of the sentence appears to be untied from each other or the study finds.

Line 313-315: I haven’t seen any signal of plasticity or adaptability, the last not even close to being tested during the study.

Line 315-323: I could not understand the link between the previous sentence and this part of the text.

Line 324-325: Strange sentence arrangement. Furthermore, the entire paragraph could be easily deleted.

I believe that the discussion needs to be rewritten and based only on the study finds, for example, what are the negative effects of being less aggressive? What is the physiological downside of change such behaviour? Could the behaviour change be beneficial to the individual, population or community?

Jumping to the conclusion which I found extremely shallow, I would suggest the authors, to make deep changes and finalize the text with the most important take away message from the results.

Reviewer #2: This study deals with understanding the behavioral changes of Stegastes fuscus under warm water conditions.

The results from the manuscript are overall clear, and the statistical analyses are good. However there are several changes that need to be completed before the manuscript is published. The most pressing problem is that the article is filled with multiple errors in grammar and syntax that make the manuscript somewhat difficult to follow. I would respectfully encourage the authors to use either a software (Grammarly for example) or the help from a colleague to improve the writing. Here I am only presenting detailed comments for the Introduction, but a thorough revision should be done for the whole manuscript.

One of the concerns that I have with the introduction is that the authors mention multiple times that coral reef fishes are resilient to climate change, or that some species are (including S. fuscus) are not thermally sensitive. I think the authors should be careful about this sort of claims, as even when fishes don't die immediately after a acute warming (like corals), they do suffer detrimental effects in their aerobic metabolism, reproduction, developmental rates and behavior. I think a better approach to modify the introduction is to discuss the detrimental effects on fishes first, and then also discuss how increased temperatures will affect coral cover and reef structure (But avoid comparing corals to fishes), and how barren environments can affect the behavior in the end.

The questions for the manuscript are not clearly presented, thus it might be better to change the small summary at the end of the introduction (Lines 85-90) with the main research questions and predictions of what the authors expect to find.

There is also information missing from some of the methods. The experimental design is a bit unclear, line 113 fore example, was there one fish per tank? Further there was no mention as to what was the temperature on the experimental tank where the behavior was recorded? was it 28C or 34C? This is important as some of the fish from the experiment might be responding to shock, rather than the experimental conditions. There is also no mention of the software packages used for the statistical analyses (was it all done in R?).

The discussion is unfortunately not very clear, as many ideas are repeated multiple times, making it difficult to follow. I must suggest the authors to revise this section extensively. The authors should mention in the discussion that the experimental temperatures are well within the range that these species experience today, and that conditions on those tidepools might be much warmer than 34C by the end of century. Another limitation is that the authors don't discuss the specific effect of fish stress when being in a barren environment for along time. Please include in the discussion the manuscripts by:

- von Krogh, K., Sørensen, C., Nilsson, G. E., & Øverli, Ø. (2010). Forebrain cell proliferation, behavior, and physiology of zebrafish, Danio rerio, kept in enriched or barren environments. Physiology & behavior, 101(1), 32-39.

- Näslund, J., & Johnsson, J. I. (2016). Environmental enrichment for fish in captive environments: effects of physical structures and substrates. Fish and Fisheries, 17(1), 1-30.

There are some ideas in the discussion that are not well stated. For example, lines 359-361 suggest that changes in temperature lead to changes in salinity and O2, which leads to coral mortality. The main cause of bleaching is temperature increase and the loss of the association between symbionts and corals.

Finally, the authors could make some of the fish videos available on Youtube or other video sharing website, so colleagues can see the experiments. This would be a nice addition to the manuscript.

Minor revisions:

Line 39: The more drastic change in pH is caused by the increase of Partial pressure of atmospheric CO2. Thus I don't know how relevant pH is in this statement.

Line 44: Underwater is one word

Line 50: replace "species" with "organisms", since you are not talking about a specific group.

Line 55-58: This line is a bit confusing, please consider reorganizing these ideas.

Line 59: I must encourage the authors to revise this. Coral reef fishes are very sensitive to changes in water temperature. Rephrase this to "Previous studies suggest that changes in behavior of coral reef fishes could be associated to fluctuations in environmental conditions".

Line 61: Replace "in this respect" with "Thus,"

Line 66: is this diel migrations, or actual long range movements?

Line 67: Again I think it's tricky to suggest that some fishes are not affected by temperature. A much better approach could be just describing the distribution of the dusky damselfish, rather than speculating that it's not thermally sensitive.

Line 81: "may increas 2-4C on average,"

Lines 85-90: This section should have the main questions of the manuscript. At the moment it is more like a summary of the paper. Please consider re-structuring this section.

Line 104: replace "captures" with "samplings"

Line 159: Please check the numbers of the Tanks. Is this the correct name of the treatments?

Lines 313-315: I don't think adaptability was measured here, since the effect that was measured was plasticity.

Lines 344-349: This section is confusing, as they both say very similar statements, but in one sentence it is temperature, but in the other it is CO2 and O2. Please edit this section for clarity.

Lines 359: "such as salinity"

Line 359-361: This doesn't seem right. Please edit for clarity.

Reviewer #3: This ms aims to provide data on damselfish behavior using a mirror exposure and mimicking the presence of conspecifics in lab experiments facing high water temperature (34oC) and combined effect of lack of complexity. The Dusky damselfish is endemic to the Brazilian coast and since distribution occurs from tropical to subtropical latitudes (5oS to 27oS), it is considered a thermotolerant species. Damselfishes are claimed to be key species as they can enhance diversity, PPL rates and biomass of algae inside their territories, thus they can function as good indicators. As a result, the authors concluded that temperature and complexity combined affected mobility and aggressiveness. Warmer temperatures caused a decrease in displays like threats and attacks and an increase in vigilance and immobility. Low complexity treatments caused more aggressive displays than those in complex scenarios. The project idea is interesting for all those working in marine environments, namely reef systems, to understand how organisms will cope with environmental changes. Although the results are compatible with expectations I suggest authors to engage in additional efforts to convince readers that those experiments are worthy for publication. I see that many parts of the ms need more work and reference support, including the introduction and discussion. Especially in the discussion, there are many speculations that make the text fragile based on what was tested and proposed in lab. The experimental design needs much more details on the ms.

General comments!

- The Dusky damselfish was considered by the authors as thermotolerant species, as being widespread along the Brazilian coast, as so, why this species would be a good indicator for changes in temperature or another impact related to climate change? If in the natural habitats, tide pools, this species can cope with 36oC, how to support this species as good to lab experiments on this topic instead of a less resistant species?

- The hypotheses need to be clarified; the design is not clear for aggressiveness.

- Damselfishes are very resistant to life in aquarium and definitely good for experiments in lab. Their natural diet includes algae and a little bit of live, animal material, depending on species. However, in captivity they can accept many different ratios. That said, if there is possibility of running experiments on the natural habitat, where different regimes of temperature can be managed, and fishes are health with a natural diet, why not have these comparisons done?

- Two samples of twelve fishes (n=24) were reported, but disease events and discarded fishes were also mentioned, how many fishes were finally used to run each treatment and how was this managed with statistical analysis? How 36 experiments were run and only 24 fishes were available? It will be of great help see n=samples number above graphics!!

- Each 12 tanks had the same circulation system, so there are two blocks missing in the analysis.

- What is a control for each treatment?

- Q3 and Q4 seem to be same thing. It seems that from a 45 degrees of the mirror, if one are far from the base, it can see its reflection.

- How a fish was chose for the test tank? Are they take back for the same block and tank?

- Did the Greenhouse-Geisser correction effect was tested anyway?

- I am not convinced that have time out of the analysis is corrected! Fish would count as a random factor. The Q comparisons seem not necessary.

- How the experiment has 96 degrees of freedom with only 36 replicates?

- The final message, “mitigating the effects of global warming", is tricky, but not of general interested for scientific public working in reef systems. For that, I suggest based on what the results show, to indicate further experiments needed to understand effects of temperature on reef organisms including fishes. Managers need to know how stressors will affect the normal functions of natural systems, especially those mediated by species. But also important is how they can implement management or conservation strategies that could mitigate global change effects.

Line 40 – characterized by HIGH species diversity;

Line 41 - Most of tropical reefs do not go through high temperature fluctuactions, so this generalization need to be rephrased;

Line 42 – overtheir;

Line 47 – For "feeding needs", you meant nutritional requeriments?

Line 56 - Need rephrasing. Typical reported phase shifts events include domination of algae over corals. Corals are important framework of tropical reefs. Algae mortality is another interesting event but not usually reported for tropical reefs. Please, try to rephrase this sentence in order to a better understand of what process you really want readers to pay attention;

Line 65 – reproduction PATTERNS, COULD influence;

Line 70 - This affirmative of no physiological responses in face of a high latitudinal distribution needs a reference support, it is just too speculative, but an interesting topic to be further investigated;

Line 75 – Need a reference in studies considering the Dusky Damselfish;

Line 77 – Needs a proper reference;

Line 84 – Energy=nutrients for damselfishes came from their turf matrix in territories, which includes algae, detritus and associated cryptofauna. All these itens can be part fo their diets pending on species. How these food sources will be affected by rise of temperature? Feeding rates clearly are affected by temperature. Territorial defence is a density dependant process which needs to be discussed considering the density of conspecifics and heterospecifics;

Line 309 – One important aspect not evaluated by authors is feeding rates. Feeding rates is an important measure to understand patterns of energy flux throughout the food web. Lab experiments can also be important to test how feeding rates will be affected by temperature. Said that, the discussion need more on that issue;

Line 343 – Physiological displays = vigilance and immobility?? These are in fact behavioural aspects!!

Line 359 – as salinity;

Line 382 – How rise in water temperature may decrease food availability? This assumption need more details.

6. PLOS authors have the option to publish the peer review history of their article (what does this mean?). If published, this will include your full peer review and any attached files.

Reviewer #1: No

Reviewer #2: No

Reviewer #3: No

---

## [Author Response · Author response to Decision Letter 0]

17 Dec 2019

Rebuttal letter 

Reviewer #1: General comments:

Dear reviewer #1, thankyou for your time and comments when revising our manuscript. In fact, the article approaches only one fish species, as it was a study developed during the master degree enrollment of the first author. Our aim was not to study the effects of warming on the whole ecosystem, as it would have taken several years, but however, we could address an up to date topic by responding a simple question of behavioral change, and thus we focused in the most characteristic behavior of an endemic fish species: aggression. 

Regarding your concerns about the text, you will notice that we have accepted all suggestions from you and the other 2 reviewers, and the final version was modified and improved. The specific comment you have pointed are approached bellow:

Specific comments:

Abstract:

Line 18: The author's statement “Due to the high anthropogenic impact of temperature, the pH, oxygen and structural complexity of many environments has changed.” sounds wrong. I would rephrase it. Warming is caused by the increase in atmospheric CO2 concentration, and the way that warming regulates pH could be somehow questionable, once that drops in pH (ocean acidification) is also directly associated with elevated atmospheric CO2 and primary production. The first sentence was changed to be more adequate to the context of the article 

Line 24: This is unnecessary information “Fish behavior was recorded for 5 min before 25 and 5 min after mirror exposure.”- sentence was removed 

Line 28-30: Motherhood statement that does not move the field forward and also does not match with the study finds. Sentence was removed and a new final phrase was added

Introduction:

The introduction was revised and many parts were altered, as suggested. 

Line 35-58: The first two paragraphs could be easily mutated to a single paragraph. Other reviewers have suggested changes to these paragraphs; thus it was almost completely changed. 

Line 80-91: The hypothesis tested is based on assumptions not evaluated during the experiment which in my opinion is the biggest flaw in the study. Climate change is known for affecting species with different strength (some species may have suffered stronger physiological setbacks than others), such negative effects could be diminished or buffered by environmental complexity (see Goldenberg et al. 2018) and species interactions. Additionally, behaviour changes could be a simple and straightforward strategy to diminish physiological impairments or even take a chance to boost performance (Ferreira et al. 2018). The text was rewritten and the hypothesis is clearly presented.

Materials and Methods:

Line 103: the sample size appears to be wrong by my calculations you collected 36 fish. Am I wrong? We have included a statement presenting the reason why sample size differs. In fact, we have sampled 48 fish, however, due to sampling with a net, transportation to the lab and the changing in holding conditions, we have to exclude 12 fish. Some of them were excluded because of injuries during the sampling, other became sick at the lab (probably because these fish were not in the better health conditions when were captured and the stress may have declined their conditions, and other were excluded because they stopped feeding at the lab). Thus, final total number was 36 fish. 

Results:

First, due to low replication (sampling size vary from 6 to 12), I would appreciate seeing individual data points in the figures. Changed as requested.

Second, Figure 3, in my opinion, is not necessary. We disagree that figure 3 should be deleted. It shows area occupancy before and after the mirror exposure and the figure indicates that fish were not using the closet area to the mirror before (Q4), but as soon as the mirror is shown, fish respond to it and occupied area Q1. Although it was not affected by temperature and complexity, the figure shows that fish still respond to the “intruder”. 

Figure 4, does not necessarily help since behaviour is not been affected by habitat complexity (left panel) and temperature (right panel) appears to have only a mild effect on fish behaviour, and as I said before, complexity buffer warming effects (Goldenberg et al. 2018). Again, that figure shown behavioral change and as other reviewers did not mention anything related to excluding these figures, we decided to maintain them. 

I do have a question. During the behaviour test, the authors took feeding into account. However, is not clear in the experiment design how much food was available or how the authors made it available. Was it turf algae that grew in the experimental setup or it was food pallets? Was the food available in the same amount for all fish? Note that it could have serious implications on the results of the experiment. I would expect that fish in low complexity and higher temperatures would spend more time feeding (based on Metabolic Ecology Theory by Brown et al. 2004), or gardening (Ferreira et al. 2018, to avoid the overgrown of high nutritious algae by weedy ones in damselfish territories). Yet, the authors do not give any indication of such responses. Indeed, we have not offered food during the behavioral test (10 min in the test tank). The dusky damselfish is known to be a gardener in the natural environment and the bottom nibbling is a innate behavior that was observed even when no food was available. Thus, we recorded the behavior as it was part of the repertory, but we changed the nomenclature to “substrate nibble” instead of “feeding”. However, we in fact observed that fish reduced feeding over the 30 days period, mainly in the barren warmer condition, but as we have not consistently recorded the amount of food consumed, we did not include this information in the MS.

Discussion:

The discussion was entirely revised and most of it re-written.

Line 313-315: I haven’t seen any signal of plasticity or adaptability, the last not even close to being tested during the study. Adaptability was removed from the sentence and we added a citation, as this information was not produced by our data, but is a result of other authors study 

Line 315-323: I could not understand the link between the previous sentence and this part of the text. The sentence was removed and a novel paragraph was built

Line 324-325: Strange sentence arrangement. Furthermore, the entire paragraph could be easily deleted. As the text was re-written, the pointed phrase was removed 

I believe that the discussion needs to be rewritten and based only on the study finds, for example, what are the negative effects of being less aggressive? What is the physiological downside of change such behaviour? Could the behaviour change be beneficial to the individual, population or community? We agree with the reviewer and the discussion was almost entirely re written. 

Jumping to the conclusion which I found extremely shallow, I would suggest the authors, to make deep changes and finalize the text with the most important take away message from the results. The last part of the discussion is more robust in this new version.

Reviewer #2: 

Dear reviewer #2, we are very glad with your comment and suggestions to improve our text. We have accepted all suggestions and you will find almost completely re-written text. We agree that we have been using a confusing statement when we say that S. fuscus is tolerant to thermic stress. In fact, these fish have more wide tolerance range, however, temperature changes affect them and cause lots of detrimental physiological and behavioral responses. We have changed our approach and included some novel statements to our MS. 

The questions for the manuscript are not clearly presented, thus it might be better to change the small summary at the end of the introduction (Lines 85-90) with the main research questions and predictions of what the authors expect to find. We understand the point. We have changed the text and avoided characterizing the fish as thermotolerant, presenting the area and temperature range it occurs. From lines 85 to 90, it was completely rewritten and a clear hypothesis is presented. 

There is also information missing from some of the methods. The experimental design is a bit unclear, line 113 fore example, was there one fish per tank? Further there was no mention as to what was the temperature on the experimental tank where the behavior was recorded? was it 28C or 34C? This is important as some of the fish from the experiment might be responding to shock, rather than the experimental conditions. Methods section was revised and we have included missing information pointed out by the reviewer. In fact, fish from each holding temperature was also tested in the same temperature, i. e. fish from 28 was tested in 28 degrees and fish from 34 was tested in 34 degrees. Also, fish were held in visual isolation from neighbours to avoid physical combat and injuries, but they were kept with chemical communication through water exchange. 

There is also no mention of the software packages used for the statistical analyses (was it all done in R?) Univariate analyses (two-way and repeated measures ANOVA tests) were performed in the software Systat 12 and the multivariate procedures (PERMANOVA and SIMPER tests) in the software Primer 6 with PERMANOVA add-on. This information are now mentioned in the revised version of the manuscript. 

The discussion is unfortunately not very clear, as many ideas are repeated multiple times, making it difficult to follow. I must suggest the authors to revise this section extensively. The authors should mention in the discussion that the experimental temperatures are well within the range that these species experience today, and that conditions on those tidepools might be much warmer than 34C by the end of century. 

Another limitation is that the authors don't discuss the specific effect of fish stress when being in a barren environment for along time. Please include in the discussion the manuscripts by:

- von Krogh, K., Sørensen, C., Nilsson, G. E., & Øverli, Ø. (2010). Forebrain cell proliferation, behavior, and physiology of zebrafish, Danio rerio, kept in enriched or barren environments. Physiology & behavior, 101(1), 32-39.

- Näslund, J., & Johnsson, J. I. (2016). Environmental enrichment for fish in captive environments: effects of physical structures and substrates. Fish and Fisheries, 17(1), 1-30.

There are some ideas in the discussion that are not well stated. For example, lines 359-361 suggest that changes in temperature lead to changes in salinity and O2, which leads to coral mortality. The main cause of bleaching is temperature increase and the loss of the association between symbionts and corals.

Finally, the authors could make some of the fish videos available on Youtube or other video sharing website, so colleagues can see the experiments. This would be a nice addition to the manuscript.. We accept the suggestion and we have chosen one of the videos to make available on youtube. The address is cited on the MS.

Minor revisions:

Line 39: The more drastic change in pH is caused by the increase of Partial pressure of atmospheric CO2. Thus I don't know how relevant pH is in this statement. We removed mentions to pH

Line 44: Underwater is one word corrected

Line 50: replace "species" with "organisms", since you are not talking about a specific group. corrected

Line 55-58: This line is a bit confusing, please consider reorganizing these ideas. Re-written

Line 59: I must encourage the authors to revise this. Coral reef fishes are very sensitive to changes in water temperature. Rephrase this to "Previous studies suggest that changes in behavior of coral reef fishes could be associated to fluctuations in environmental conditions". Done

Line 61: Replace "in this respect" with "Thus," corrected

Line 66: is this diel migrations, or actual long range movements? We removed mentions to migration

Line 67: Again I think it's tricky to suggest that some fishes are not affected by temperature. A much better approach could be just describing the distribution of the dusky damselfish, rather than speculating that it's not thermally sensitive. corrected

Line 81: "may increas 2-4C on average," corrected

Lines 85-90: This section should have the main questions of the manuscript. At the moment it is more like a summary of the paper. Please consider re-structuring this section. Corrected according to studies from Dr. McCormick: In this study we evaluated whether water temperature and structural complexity of habitat may be related to changes in mobility patterns, tanks occupation and behavioral profile of Stegastes fuscus. For this, we subject the animals to classic mirror test and observe if animals kept at high temperature and barren conditions present significant differences in the mentioned variables when compared with group kept in high temperature and enriched habitat (similar to natural habitat). As an increase in temperature raises the metabolic rate of fish and this metabolic alteration promote direct influences in behavior of animals expected that high temperature groups and barren conditions show the lowest results. 

Line 104: replace "captures" with "samplings" corrected

Line 159: Please check the numbers of the Tanks. Is this the correct name of the treatments? Names and numbers are correct

Lines 313-315: I don't think adaptability was measured here, since the effect that was measured was plasticity. ok, removed

Lines 344-349: This section is confusing, as they both say very similar statements, but in one sentence it is temperature, but in the other it is CO2 and O2. Please edit this section for clarity. It was re-written

Lines 359: "such as salinity" corrected

Line 359-361: This doesn't seem right. Please edit for clarity. corrected

Reviewer #3: 

Dear reviewer #3, thankyou very much for considering our study and suggesting nice modifications to make it clearer and better. We are very thankful for your time and suggestions. We have accepted all of them and most parts of the MS (mainly introduction and discussion) were completely rewritten and new references were added. Please, find the new version of the MS with all suggestions from the 3 reviewers. 

General comments!

- The Dusky damselfish was considered by the authors as thermotolerant species, as being widespread along the Brazilian coast, as so, why this species would be a good indicator for changes in temperature or another impact related to climate change? If in the natural habitats, tide pools, this species can cope with 36oC, how to support this species as good to lab experiments on this topic instead of a less resistant species? Two main reason made us focus on S. fuscus: 1 - Easy collection (high abundance) and (apparent) thermal plasticity; 2 - The species is known to tolerate high temperatures in tide pools for short periods of time (minutes / hours). The purpose of this paper is precisely to understand how animals would respond to long-term thermal stress (days), so it would not make sense to test this in another species that is known to tolerate short intervals of high temperatures

- The hypotheses need to be clarified; the design is not clear for aggressiveness. The experimental design was better explained in materials and methods. The hypothesis was more clearly described at the end of the introduction.

- Damselfishes are very resistant to life in aquarium and definitely good for experiments in lab. Their natural diet includes algae and a little bit of live, animal material, depending on species. However, in captivity they can accept many different ratios. That said, if there is possibility of running experiments on the natural habitat, where different regimes of temperature can be managed, and fishes are health with a natural diet, why not have these comparisons done? In fact, it would be excellent to compare data from nature to data from the laboratory. We plan to sample behavior in the field as we have 

been observing fish behavior both in nature and in the lab. We observed several differences, for example in relation to the territorial defense, we observed that damselfish from nature are less aggressive against conspecifics than against heterospecific fish. However, we are still analysing these results and comparing the fish defence and neighbour recognition between field and lab sampling. We believe it will be ready for submission in 2020.

- Two samples of twelve fishes (n=24) were reported, but disease events and discarded fishes were also mentioned, how many fishes were finally used to run each treatment and how was this managed with statistical analysis? How 36 experiments were run and only 24 fishes were available? It will be of great help see n=samples number above graphics!! It is better explained in the MS so that the readers will see that we have sampled 48 fish, 12 held in each closed recirculating system. However, because sampling with a net, transporting and changing holding conditions are very stressful, some fish that we believe were not in its best health became sick or stopped eating and thus we decided to exclude them from the test. Thus, we ended up with 36 fish as described in the MS.

 - Each 12 tanks had the same circulation system, so there are two blocks missing in the analysis. As I mentioned above, some fish were removed from the tests and tanks were maintained empty. We have included the details in the methods section, showing total number of animals used, number of animals composing each group and all statistical procedures. 

- What is a control for each treatment? There is not a control for each treatment. We compared barren versus enriched, thus one is the control of the other. Also, 28 degrees is the usual temperature for maintain damselfish in the laboratory, thus it was the control condition for 34 degrees. 

- Q3 and Q4 seem to be same thing. It seems that from a 45 degrees of the mirror, if one are far from the base, it can see its reflection. Yes, in fact it is the same, however, as using a single back area would double the size to be compared to the two front areas, we had to keep it separated into 2 areas. 

- How a fish was chose for the test tank? Are they take back for the same block and tank? No, each fish was used only once and was part of only one group condition. Thus, each fish was recorded and then euthanized. 

- Did the Greenhouse-Geisser correction effect was tested anyway? We used the Greenhouse-Geisser correction because the RM-ANOVA analysis returned epsilon values of 0.716 and 0.689. According to Girden (1992), when epsilon values are smaller than 0.75 the Greenhouse-Geisser correction is recommended.

- I am not convinced that have time out of the analysis is corrected! Fish would count as a random factor. The Q comparisons seem not necessary. Fish is the unit that is subjected to repeated measures both across time (before and after mirror) and on different quadrants (Q1, Q2, Q3, Q4), and therefore it is indeed a random factor. The comparison among quadrants was one of the main goals of our experimental design since the time spent in each quadrant reflect how aggressive the fish is (detailed explanation in Methods). The comparison between different times was only to evaluate eventual changes in basal behavior (before vs. after). To the best of our knowledge, there is no way of combining two levels of repeated measures in the same analyses, therefore we decided to separate it in two analyses, one for each time, using the repeated measures of the same fish on different quadrants as the response variable.

- How the experiment has 96 degrees of freedom with only 36 replicates? As each of the 36 fish was repeatedly measured four times, one in each of the four quadrants for each treatments combination, we end up with a total of 144 replicates and therefore 143 degrees of freedom (1 for each fixed factor (temperature and complexity) and their interaction (T x C); 32 for the residual of between subjects; 3 for the repeated factor (quadrants) and its interactions with fixed factors (Q x T; Q x C; Q x T x C); and 96 for the residual of within subjects.

- The final message, “mitigating the effects of global warming", is tricky, but not of general interested for scientific public working in reef systems. For that, I suggest based on what the results show, to indicate further experiments needed to understand effects of temperature on reef organisms including fishes. Managers need to know how stressors will affect the normal functions of natural systems, especially those mediated by species. But also important is how they can implement management or conservation strategies that could mitigate global change effects. We completely agree, thus we have changed our text to be clearer and to suggest next steps in this area of research.

Line 40 – characterized by HIGH species diversity; corrected

Line 41 - Most of tropical reefs do not go through high temperature fluctuactions, so this generalization need to be rephrased; corrected

Line 42 – overtheir; corrected

Line 47 – For "feeding needs", you meant nutritional requeriments? corrected

Line 56 - Need rephrasing. Typical reported phase shifts events include domination of algae over corals. Corals are important framework of tropical reefs. Algae mortality is another interesting event but not usually reported for tropical reefs. Please, try to rephrase this sentence in order to a better understand of what process you really want readers to pay attention; The text was rewritten based on suggestions. The part where we referred to algae and corals was removed. 

Line 65 – reproduction PATTERNS, COULD influence; corrected

Line 70 - This affirmative of no physiological responses in face of a high latitudinal distribution needs a reference support, it is just too speculative, but an interesting topic to be further investigated; Unfortunately, there are no studies that relate the wide geographical distribution of the species in Brazil with physiological changes found throughout its distribution. On the speculation that the species is thermotolerant, we prefer to abide by the reviewers' suggestions and remove the part describing S. fuscus as a thermotolerant species.

Line 75 – Need a reference in studies considering the Dusky Damselfish; included

Line 77 – Needs a proper reference; As we did not find reference, we decided to remove the quote from “These species contribute to energy and nutrient transfer in reef environments, as a result of their gardening ability.”

Line 84 – Energy=nutrients for damselfishes came from their turf matrix in territories, which includes algae, detritus and associated cryptofauna. All these itens can be part fo their diets pending on species. How these food sources will be affected by rise of temperature? Feeding rates clearly are affected by temperature. Territorial defence is a density dependant process which needs to be discussed considering the density of conspecifics and heterospecifics; Although we will recognize that the food items of our study species will also be affected by future increases in water temperature and possible changes in the structural complexity of the reefs, the purpose of our study was limited to analyzing the behavioral aspect of the animal. During the experiments the animals were not fed and we did not quantify the nutrient intake by the animals. We agree that territorial defense is a process dependent on the density of co and heterospecifics. Therefore, in our experiment all animals were kept in social isolation both during the acclimatization / adaptation period and during the experiment period.

Line 309 – One important aspect not evaluated by authors is feeding rates. Feeding rates is an important measure to understand patterns of energy flux throughout the food web. Lab experiments can also be important to test how feeding rates will be affected by temperature. Said that, the discussion need more on that issue; we have included a discussion regarding feeding and methabolism 

Line 343 – Physiological displays = vigilance and immobility?? These are in fact behavioural aspects!! corrected

Line 359 – as salinity; corrected

Line 382 – How rise in water temperature may decrease food availability? This assumption need more details. Increase in temperature may affect primary production and thus it can become a cascade that will end up reducing all the food chain. We have included a better approach to this topic in the discussion section.

---

## [Decision Letter · Decision Letter 1]

12 Mar 2020

PONE-D-19-24418R1

Damselfish face climate change: impact of temperature and habitat structure on agonistic behavior

PLOS ONE

Dear Ana C Luchiari,

Thank you for submitting your manuscript to PLOS ONE. After careful consideration, we feel that it has merit but does not fully meet PLOS ONE’s publication criteria as it currently stands. Therefore, we invite you to submit a revised version of the manuscript that addresses the points raised during the review process.

We would appreciate receiving your revised manuscript by April 11. To enhance the reproducibility of your results, we recommend that if applicable you deposit your laboratory protocols in protocols.io, where a protocol can be assigned its own identifier (DOI) such that it can be cited independently in the future. For instructions see: http://journals.plos.org/plosone/s/submission-guidelines#loc-laboratory-protocols

We look forward to receiving your revised manuscript.

Kind regards,

Hudson Tercio Pinheiro

Academic Editor

PLOS ONE

Additional Editor Comments (if provided):

Dear Authors,

The manuscript has been improved following the suggestion of the previous reviewers. We have now minor revisions suggested by two referees. I look forward to receiving a new version of the manuscript and consider the publication at Plos One.

Sincerely Yours,

Hudson Pinheiro

Reviewers' comments:

Reviewer's Responses to Questions

**Comments to the Author**

1. If the authors have adequately addressed your comments raised in a previous round of review and you feel that this manuscript is now acceptable for publication, you may indicate that here to bypass the “Comments to the Author” section, enter your conflict of interest statement in the “Confidential to Editor” section, and submit your "Accept" recommendation.

Reviewer #2: All comments have been addressed

Reviewer #4: All comments have been addressed

2. Is the manuscript technically sound, and do the data support the conclusions?

Reviewer #2: Yes

Reviewer #4: Yes

3. Has the statistical analysis been performed appropriately and rigorously? 

Reviewer #2: Yes

Reviewer #4: Yes

4. Have the authors made all data underlying the findings in their manuscript fully available?

Reviewer #2: Yes

Reviewer #4: Yes

5. Is the manuscript presented in an intelligible fashion and written in standard English?

Reviewer #2: Yes

Reviewer #4: Yes

6. Review Comments to the Author

Reviewer #2: The manuscript is much clearer on this version than on the original submission. For the most part the authors have done a good job replying to the concerns of the reviewers. However, I must respectfully suggest that the authors revise the manuscript before the final submission. Here are some of the sections that should be edited for clarity, but I encourage a more thorough revision of the whole manuscript.

line 37: the line of "ecological master factor" this sentence can be deleted

line 43: replace reef with reef's

line 65: replace transpire with materialize

Line 74-76: this is a very powerful conclusion that is not entirely supported by the data. I would suggest that the authors just say "we suggest that natural aggressive behavior of S. fuscus could be affected"

Line 86: change ppm to ppt (parts per thousand in english). Check the entire manuscript as this is repeated in several places

Line 104: this should be consistent throughout the manuscript, it should be 34C, not thirty-four degrees.

Lines 234-236: Please consider re-writing this line, it is a very important summary of the results, and it is a bit confusing at the moment.

Line 295: change ecological relation to "interactions"

Line 296: the species "shows"

Line 298: this has to be changed to " the rate of warming will accelerate in the near future"

Line 311: change situation for "scenario"

Line 323: you did not measure aerobic capacity or swimming speed so please consider re-shaping this section of the discussion.

Line 338: Include the abbreviation "C" for degrees Celsius.

LIne 347-329: this line is totally out of context, you could eliminate it without affecting the manuscript.

LIne 350-352: This section is confusing, please revise it.

The authors could present a picture of the fish along with Figure 1.

Figure 2 would be much better with Boxplots, this is just a suggestion and don't do the change if it's too much work.

Figure 2 and 3 need sub-headings A and B, to recognize what the bar graphs are representing. Check the journal formatting for this.

Check the format of the references, as some have different fonts, colors and styles.

Reviewer #4: Introduction

The authors have improved the MS with the previews reviews and I have few comments to clarify some points. I suggest the authors pointed out the effects of environmental changes (temperature and habitat) on reef fish behavior and what is its consequences for the coral environment. The authors should indicate the future forecast for the studied region, mainly the temperature and habitat structure projections. I suggest the authors rewrite the hypothesis since their hypothesis is only focused on temperature.

Matherial and Methods

It is not clear how did the authors calculate the swimming velocity through the Behavior recorded.

Minor Reviews

Line 38 – Change to “For water-breathing ectothermic …”

Line 323 – I suggest remove the discussion about aerobic scope since the authors did nor measured the fish metabolism and the discussion seems speculative.

7. PLOS authors have the option to publish the peer review history of their article (what does this mean?). If published, this will include your full peer review and any attached files.

Reviewer #2: No

Reviewer #4: No

---

## [Author Response · Author response to Decision Letter 1]

15 Apr 2020

Reviewer #2: General comments:

Dear reviewer, once again we thankyou for your comments on the MS. We have gone through your suggestions, which I comment below. The text was also completely revised for a native speaker. 

Specific comments:

line 37: the line of "ecological master factor" this sentence can be deleted sentence was removed

line 43: replace reef with reef's Changed as requested.

line 65: replace transpire with materialize Changed as requested.

Line 74-76: this is a very powerful conclusion that is not entirely supported by the data. I would suggest that the authors just say "we suggest that natural aggressive behavior of S. fuscus could be affected" As the other referee also asked for changes in the last part of the introduction, you will find the text is changed and adequate to what you suggested

Line 86: change ppm to ppt (parts per thousand in english). Check the entire manuscript as this is repeated inseveral places Changed as requested.

Line 104: this should be consistent throughout the manuscript, it should be 34C, not thirty-four degrees Changed as requested.

Lines 234-236: Please consider re-writing this line, it is a very important summary of the results, and it is a bit confusing at the moment We have changed the paragraph to make it clearer

Line 295: change ecological relation to "interactions" Changed as requested.

Line 296: the species "shows" Changed as requested.

Line 298: this has to be changed to " the rate of warming will accelerate in the near future" Changed as requested.

Line 311: change situation for "scenario" Changed as requested.

Line 323: you did not measure aerobic capacity or swimming speed so please consider re-shaping this section of the discussion. In fact, aerobic capacity was not evaluated, however, we did measure swimming velocity, as present in figure 2. Thus, we have changed the text to adequate.

Line 338: Include the abbreviation "C" for degrees Celsius. Changed as requested.

LIne 347-329: this line is totally out of context, you could eliminate it without affecting the manuscript. Changed as requested.

LIne 350-352: This section is confusing, please revise it. The section was rewrite

The authors could present a picture of the fish along with Figure 1. We included the fish picture as suggested

Figure 2 would be much better with Boxplots, this is just a suggestion and don't do the change if it's too much work. The first draft sent the journal was a boxplot and other referee suggested changes because it was too polluted figure, thus, we decided not to change back

Figure 2 and 3 need sub-headings A and B, to recognize what the bar graphs are representing. Check the journal formatting for this. We have described graphs A and B in the legend, as you will see

Check the format of the references, as some have different fonts, colors and styles. Changed as requested.

Reviewer #4: 

Dear reviewer #4, we are very glad about your comment to improve our text. We have accepted all suggestions, as some of them were also suggested by the other referee. We have changed our text according to your suggestions. 

Introduction was changed in some parts to make the effects of environmental changes clearer and the hypothesis was rewritten to be more adequate.

Matherial and Methods

It is not clear how did the authors calculate the swimming velocity through the Behavior recorded. We have included a statement presenting how the software deals with the video to calculate velocity. The calculations are performed on a series of frames to produce quantified measurements of the animal behavior. It is known the position of the animals for each frame of the video, and the number of frames per second. Thus, the series of frames is analyzed, and the number of frames changed in a certain time is used to estimate the animal's movement.

Minor Reviews

Line 38 – Change to “For water-breathing ectothermic …” Changed as requested.

Line 323 – I suggest remove the discussion about aerobic scope since the authors did nor measured the fish metabolism and the discussion seems speculative. The aerobic scope was removed.

---

## [Editor Report · Decision Letter 2]

18 May 2020

PONE-D-19-24418R2

Damselfish face climate change: impact of temperature and habitat structure on agonistic behavior

PLOS ONE

Dear Dr. Luchiari,

Thank you for submitting your manuscript to PLOS ONE. After careful consideration, we feel that it has merit but does not fully meet PLOS ONE’s publication criteria as it currently stands. Therefore, we invite you to submit a revised version of the manuscript that addresses the points raised during the review process.

We would appreciate receiving your revised manuscript by Jul 02 2020 11:59PM. To enhance the reproducibility of your results, we recommend that if applicable you deposit your laboratory protocols in protocols.io, where a protocol can be assigned its own identifier (DOI) such that it can be cited independently in the future. For instructions see: http://journals.plos.org/plosone/s/submission-guidelines#loc-laboratory-protocols

We look forward to receiving your revised manuscript.

Kind regards,

Hudson

Academic Editor

PLOS ONE

Additional Editor Comments (if provided):

Dear Dr Luchiari and co-authors,

I am pleased to accept your manuscript for publication in PLOS ONE. I have only one additional request. Since the whole experiment was recorded in video, I think it would be interesting to have a short video clip showing methods and behaviors (results) associated with the manuscript, what would bring more visibility to the paper and the journal through media. Is it possible to prepare this video and add a link in the methods and results sections?

Best Regards

Hudson
---

## [Author Response · Author response to Decision Letter 2]

2 Jun 2020

Dear editor, the video required was prepared and is presented at the end of the methods section. Please check the video at https://www.youtube.com/watch?v=wz_aOkiunOA&feature=youtu.be

---

## [Editor Report · Decision Letter 3]

16 Jun 2020

Damselfish face climate change: impact of temperature and habitat structure on agonistic behavior

PONE-D-19-24418R3

Dear Dr. Luchiari,

We’re pleased to inform you that your manuscript has been judged scientifically suitable for publication and will be formally accepted for publication once it meets all outstanding technical requirements.

Kind regards,

Hudson Tercio Pinheiro

Academic Editor

PLOS ONE

---

## [Editor Report · Acceptance letter]

19 Jun 2020

PONE-D-19-24418R3 

Damselfish face climate change: impact of temperature and habitat structure on agonistic behavior 

Dear Dr. Luchiari:

I'm pleased to inform you that your manuscript has been deemed suitable for publication in PLOS ONE. Congratulations! Your manuscript is now with our production department. 

Kind regards, 

on behalf of

Dr. Hudson Tercio Pinheiro 

Academic Editor

PLOS ONE